# New Approach to Chronic Back Pain Treatment: A Case Control Study

**DOI:** 10.3390/biomedicines11010073

**Published:** 2022-12-28

**Authors:** Matteo Bonetti, Dorina Lauritano, Gian Maria Ottaviani, Alessandro Fontana, Michele Frigerio, Alessio Zambello, Luigi Della Gatta, Mario Muto, Francesco Carinci

**Affiliations:** 1Department of Neuroradiology, Istituto Clinico Città di Brescia, 25128 Brescia, Italy; 2Department of Translational Medicine, University of Ferrara, 44121 Ferrara, Italy; 3Emergency and Urgency Department, Spedali Civili di Brescia, 25123 Brescia, Italy; 4Department of Neuroradiology, Spedali Civili di Brescia, 25123 Brescia, Italy; 5Anesthesia and Pain Therapy Service, Casa di Cura Borghi, 21020 Brebbia, Italy; 6Department of Neuroradiology, Ospedale Cardarelli, 80131 Napoli, Italy

**Keywords:** intraforaminal infiltration, steroid, oxygen-ozone, low-back pain, alpha lipoic acid, palmitoylethanolamide, myrrh

## Abstract

Background and objective: Our study compares the clinical outcome of chronic low back pain present for over six months treated with alpha-lipoic acid (ALA) + palmitoylethanolamide (PEA) and myrrh and periradicular infiltrations of oxygen-ozone under CT guide to periradicular steroidal infiltrations in a short (one week), medium (three months) and long-term period (six months). Methods: We enrolled 246 patients (Group A) with low back pain treated with periradicular infiltrations of oxygen-ozone under CT guide combined with 800 mg/day of ALA + 600 mg/day of PEA + 200 mg/day of myrrh orally. Group B consisted of 176 patients with low back pain treated with periradicular infiltrations of steroids. Patients were clinically monitored one week after the end of treatment, at three months, and at six months using a modified version of McNab’s method. Results: In Group A, the one-week clinical follow-up registered a complete remission of painful symptoms in 206 patients (83.7%), and this manifestation remained optimal in 191 patients at the three-month follow-up (77.6%) and in 178 at six months (72.3%). While the results were satisfactory in 28 patients (10.9%) at one week, 32 (13%) in the medium term, and 41 (16.6%) in the long term, non-significant results were found in 12 patients in the control at one week (4.6%), in 23 at three months (9.3%) and in 27 at six months (10.9%). In Group B, at the short-term follow-up we obtained an excellent clinical result in 103 patients (80.5%), while at three months 85 patients reported the persistence of clinical benefit (66.4%) and at six months, 72 (56.2%) reported the same result. The result was rated satisfactory in 11 (8.5%) and poor in 4 (3%). At the three-month follow-up, 23 (18%) reported a satisfactory result, and 20 (15.6%) had a poor result. At six months, 24 (18.8%) reported the persistence of a satisfactory result while for 32 the result was poor (25%). Conclusion: The results highlight how the treatment associated with ozone therapy and oral administration of alpha-lipoic acid + palmitoylethanolamide and myrrh can be considered a valid alternative to common therapeutic approaches in the treatment of chronic low back pain.

## 1. Introduction

Low back pain, with or without the involvement of the sciatic nerve, affects the majority of the population at least once in their life and is the main cause of lost workdays, with a strong impact on national health expenditure. In particular, over 70% of people in developed countries develop low back pain at a certain point in their life. This condition usually improves within two weeks, however, about 10% remain off work and about 20% experience pain after one year. Between 15% and 45% of adults experience low back pain each year, and 5% of people refer to hospital care lamenting a new acute episode [1,2]. Until thirty years ago, surgery was the treatment of choice, but now conservative measures are preferred following unsatisfactory surgical results [3,4,5]. Among the techniques adopted in the last decade for the treatment of sciatic nerve pain caused by disc herniation or by diseases of the non-discal spine (osteophytosis, spondylolysis, facet joint syndrome, etc.) there are two types of intervention: intraforaminal injection of a mixture of oxygen-ozone (O_2_-O_3_) gas [5,6,7,8] and periradicular infiltration of a steroid [9,10,11,12,13,14]. Both methods have produced encouraging results [15,16,17,18,19].

When injected, the highly oxidizing oxygen-ozone mixture causes a controlled “micro-oxidation” that produces a modulation of the cellular antioxidant system and the inflammation system. This mixture provides many benefits, such as good tissue diffusion, and anti-inflammatory, analgesic, and anti-edema properties [20,21].

It has been demonstrated that the mixture of oxygen and ozone gas reacts with the amino acids and carbohydrates that make up the nucleus pulposus of the intervertebral disc, allowing the volume reduction of the disc itself. In the case of compression of the nerve roots following a discopathy, the oxygen-ozone mixture can determine a resolution to the clinical situation. In particular, oxygen-ozone therapy produces various beneficial effects: it reduces the inflammatory process and related symptoms such as pain, promotes the elimination of non-vital tissue in the intervertebral disc thus decreasing its volume and compression of the spinal nerve root, and promotes the elimination of harmful substances and toxins thanks to the stimulating action on the microcirculation [21,22,23].

Recent reviews—such as De Sire et al. [23], Sconza et al. [24], and Hidalgo-Tallón et al. [21]—confirm these properties, also underline the necessity of investigating further the means and methods of treating with the oxygen-ozone mixture. Hidalgo-Tallón et al. [21] conclude that a more accurate standardization in treatment protocols is needed.

The possibility to orally combine alpha-lipoic acid (ALA) + palmitoylethanolamide (PEA) and myrrh with guided oxygen-ozone CT infiltration has been documented to be effective in further improving the clinical outcome. In a recent study, this association has been assessed and the authors concluded that it “can be considered an excellent therapeutic solution, improving the good final clinical result with better control of symptoms, particularly in the first phase of the disease” [25]. However, in that study, the necessity of further investigating the therapeutic effects of alpha-lipoic acid (ALA) + palmitoylethanolamide (PEA) and myrrh with guided oxygen-ozone CT infiltration was underlined.

In light of these considerations, we, therefore, decided to further evaluate the therapeutic results in a group of 246 patients treated with CT-guided infiltrations of oxygen-ozone associated with alpha-lipoic acid (ALA) + palmitoylethanolamide (PEA) and myrrh. What led us to associate the therapies in this study was the need to investigate in more depth the enhancement that they bring to the analgesic and anti-inflammatory actions of oxygen-ozone therapy alone, together with the greater maintenance of its clinical results over time, as suggested by the literature [25].

This association was thought of in light of the known analgesic action of ALA, which is a natural sulfur compound produced in small concentrations by all cells. ALA is, in fact, a key compound in some mitochondrial enzyme complexes (pyruvate dehydrogenase and ketoglutarate dehydrogenase), which play a central role in oxidative metabolism, and it can reduce oxidative stress by preventing oxygen-free radical damage [26,27,28,29,30,31,32]. Antioxidants can have functionality in aqueous or fatty tissues. ALA is an antioxidant that exerts its antioxidant function in both of these tissues. This property gives thioctic acid a broad spectrum of antioxidant action [33,34,35,36].

Myrrh extract blocks proteins involved in the inflammatory process such as Cox and inhibits the formation of NO, ROS, TNF-α, PGE2, NF-kB, and MAPK. Clinical studies indicate that myrrh extract is also able to induce significant improvements in osteoarthritis [37,38,39,40]. MyrLiq^®^ myrrh is a dry extract of Commiphora myrrh gummy resins with a high content of bioactive furanodienes. This extract—obtained through a licensing process—allows to preserve all the properties of the original raw material.

Palmitoylethanolamide (PEA) is a natural compound rich in fatty acids which, in the body, acts as a biological modulator, favoring the physiological response of the tissues [41,42,43,44,45,46,47,48,49,50].

It has been widely reported in the literature that oxygen-ozone is more effective than therapy through sole steroid administration [8]. However, to our knowledge, no other study aimed to assess and compare treatment with alpha-lipoic acid (ALA) + palmitoylethanolamide (PEA) + myrrh + guided oxygen-ozone CT infiltration and treatment with periradicular infiltrations of steroid.

In this study, our aim was to evaluate the efficacy of these two therapeutic approaches and to analyze and compare their effect over time, in short (one week), medium (3 months), and long (6 months) terms.

## 2. Methods

From March 2019 to March 2022, we enrolled 246 patients (Group A) (138 male and 108 female, aged between 52 and 81, average age 68) with chronic low back pain treated with periradicular infiltrations of oxygen-ozone under CT guide combined with 800 mg/day of ALA + 600 mg/day of PEA + 200 mg/day of myrrh orally, after providing informed consent. Patients took 2 tablets/day for 30 days.

In our sample, we included exclusively patients that had undergone a computed tomography (CT) or magnetic resonance (MR) scan. Patients manifested chronic low back pain and sciatica, which was unilateral or irradiating along the innervation territories of L3 (8 patients), L4 (71 patients), L5 (103 patients), and S1 (64 patients).

Patients with bilateral lower back and sciatic nerve pain, and those with electromyographic features of neurogenic injury and/or denervation, were excluded and advised to seek neurosurgical treatment.

Duration of symptoms varied from six to 20 months. We also enrolled 176 patients (Group B) (94 male and 82 women, aged between 54 and 76, average age 67) with low back pain, treating them with periradicular infiltrations of steroids. In this group, patients manifested chronic low back pain and sciatica, which was unilateral or irradiating along the innervation territories of L3 (6 patients), L4 (59 patients), L5 (63 patients), and S1 (48 patients), and also—in this group of patients—painful symptoms were present for 6 to 20 months.

### 2.1. Infiltration Technique

The injection site was disinfected, and local anesthesia was applied using an ethyl chloride spray in all patients. Infiltrations were completed by specialist neuroradiologists. The puncture site was identified by a CT scan and marked on the patient’s skin.

The distance from this point to the foramen was subsequently measured.

A 22 G Terumo needle (a 9 cm needle was typically used, but longer needles were occasionally adopted depending on the size of the patient) was positioned 2–3 mm from the foraminal region, close to the ganglion of the affected nerve root. A CT scan was then repeated to check the correct needle placement. This procedure was adopted for infiltrations of both O_2_-O_3_ and steroids.

Infiltration of 2 cc (80 mg) Depomedrol^®^ was completed without contrast medium, paying special attention to the speed of infiltration. The procedure was completed slowly (over 1–2 min), to avoid the reflux of steroids along the needle trajectory. Patients with facet joint syndrome received injections into the surrounding joint capsule.

The O_2_-O_3_ was infiltrated by injecting 3 cc of the gas mixture at 25 µg/mL close to the neural foramen, then retracting the needle a few millimeters and injecting another 5 cc of the mixture to involve the facet joint region. CT scans were then used to check the correct distribution of the gas mixture in the foramen and facet joint. All treatments were carried out using equipment with photometric detection of the ozone concentration in the gas mixture (the device automatically corrects the deviation of concentration that exists when the syringe withdrawal is carried out), with constant pressure during the ozone intake operation. Total non-toxicity has been guaranteed, since the titanium, teflon, glass, and silicon necessary for therapeutic practice, which are inert to ozone (Maxi Ozon Active International produced by Medica srl or Alnitec Device both CE mark class 2A) (Figure 1 and Figure 2A,B).

Additionally, clinical outcome (see tables) was assessed in all patients by short (one week), medium (three months), and long-term (six months) follow-up using a modified version of McNab’s method:(a)excellent: resolution of pain, and return to normal activity carried out prior to pain onset;(b)good or satisfactory: more than 50% reduction of pain;(c)mediocre or poor: partial reduction of pain below 70%.

### 2.2. Statistical Analysis

To quantify the time-related change in the proportion of subjects with a poor, good and excellent outcome between the O_2_-O_3_-ALA and the steroid groups, we used an ordinal logistic regression model (Rabe-Hesketh S, Skrondal A. Multilevel and longitudinal modeling using Stata. Volume II: Categorical Responses, Counts, and Survival. College Station, TX: Stata Press; 2021). This model considered the outcome (ordinal: poor, good, and excellent) as the response variable and treatment (categorical: 0 = steroid; 1 = O_2_-O_3_-ALA), occasion (categorical: 0 = visit II; 1 = visit III; 2 = 60 days), and a treatment X occasion (categorical X categorical) interaction as predictors. The model employed cluster confidence intervals to take into account repeated measures. Using this model, the difference in the proportion of patients with a poor, good and excellent outcome between O_2_-O_3_-ALA and steroid groups at the different occasions was calculated using a within-time between-group contrast with Bonferroni correction for nine groups (1 difference X 3 outcome levels X 3 occasions). Statistical analysis was performed using Stata 17.0 (Stata Corporation, College Station, TX, USA).

Figure 3 plots the time-related changes in the proportion of patients with poor, good, and excellent outcomes in the O_2_-O_3_-ALA and steroid groups.

Figure 4 plots the between-group (O_2_-O_3_-ALA minus steroid) difference in the proportion of patients with poor, good, and excellent outcomes during the study.

## 3. Results

In Group A, the short-term clinical follow-up (one week) produced an excellent result in 206 patients (83.7%) with complete remission of painful symptoms, which remained optimal in 191 patients at the three-month follow-up (77.6%) and in 178 to 6 months (72.3%). While the results were satisfactory in 28 patients (10.9%) at one week, 32 (13%) in the medium term, and 41 (16.6%) in the long term, non-significant results were found in 12 patients in the control at one week (4.6%), in 23 at three months (9.3%) and in 27 at six months (10.9%).

In Group B, at the short-term follow-up, we obtained an excellent clinical result in 103 patients (80.5%), while at 3 months 85 patients reported the persistence of clinical benefit (66.4%) and at six months 72 (56, 2%) reported the same result. The result was rated satisfactory in 11 (8.5%) and poor in 4 (3%). At the 3-month follow-up, 23 (18%) reported a satisfactory result and 20 (15.6%) a poor result. At 6 months, 24 (18.8) reported the persistence of a satisfactory result while for 32 the result was poor (25%).

At short-term follow-up, most of the patients in both groups experienced a complete remission of pain (80.5% of patients treated with Depomedrol^®^ and 83.7% of patients treated with the O_2_-O_3_ mixture associated with ALA + PEA+ Myrrh). Thus, a good result was recorded regardless of the administered treatment, with effects that did not reach statistical significance (Figure 3 and Figure 4).

At medium-term follow-up (3 months), 77.6% of the patients in Group A and 66.4% of the patients in Group B remained pain-free.

At long-term follow-up (6 months), 72.3% of Group A treated with ozone and ALA + PEA + Myrrh and 56.2% of group B treated with Depomedrol^®^ deemed the clinical outcome to be excellent.

Therefore, the ozone treatment associated with ALA + PEA + Myrrh was decidedly preferred in the medium and long term as a therapeutic result compared to steroid infiltrations.

## 4. Discussion

In this observational study, we found that treatment with O_2_-O_3_-ALA as compared to steroids was associated with a higher number of excellent results at 60 days (+17%, *p* = 0.0216) corresponding to −10% of poor cases (*p* = 0.0036) and −7% of good cases, (*p* = 0.0079).

The short-term outcome in our series was similar in both groups of patients, regardless of the type of treatment administered. The basis of this pain relief, produced by both steroid and ozone infiltration, appears to reflect the origin of nerve root pain. Zennaro and others claim that low back pain and sciatica have a neuritic origin. In fact, episodes of back pain and sciatica are linked to factors not strictly connected to the mechanical compression of the nerve root, but caused by an aspecific inflammatory reaction to the auto-antigens of the mucopolysaccharide matrix located on the disc surface, and exposure to contact with the immune system by migration of the disc nucleus beyond the natural barrier of the annulus fibrosus [9]. The inflammatory reaction is linked to the release of lytic enzymes, such as E2 prostaglandins and A2 phospholipase, present in the peridiscal epidural adipose environment in quantities a hundred-fold higher in disc herniation than in patients with bulging disc alone, thereby confirming the postulated inflammatory origin of pain.

The rationale for anti-inflammatory treatment by periganglionic steroid infiltration is based on an attempt to relieve the periganglionic inflammation, by ensuring recovery of the normal ganglioneural myelin sheath, and hence nerve function at the disease site [9,10,11,12,13,14]. While the oxygen-ozone gas mixture injected proximal to the root ganglion is thought to normalize the level of cytokines and prostaglandins, increase superoxide dismutase (SOD), and minimize reactive oxidant species (ROS) and improve local periganglionic circulation with a eutrophic effect on the nerve root [1,2,3,4,5,6,7,8,9].

This was especially evident in the short-term follow-up, roughly 80% of patients reported a clear-cut improvement in symptoms, thereby demonstrating that both treatments rapidly relieve pain.

Our medium and long-term results, on the other hand, strongly favor the treatment with guided CT infiltration with oxygen-ozone associated with the administration of ALA + PEA + myrrh to the infiltration of periganglionary steroid (Figure 5A–D and Figure 6A,B).

Periganglionic steroid treatment using CT-guided paraspinal infiltration was effective in relieving root pain caused by spondylosis or herniated disc but was short-lived.

In fact, in the medium and long-term controls, the success rate with steroids and the percentage of excellent results drops significantly. In addition to this, we want to underline how steroid treatment can sometimes be burdened by complications such as arachnoiditis, meningitis, paraparesis, paraplegia, sensory disturbances, intestinal/bladder dysfunctions, headaches, and epilepsy. Although these important side effects were not found in our series, a certain percentage of the drug is still metabolized.

## 5. Conclusions

Periganglionic steroid treatment by CT-guided paraspinal infiltration is an effective tool for relieving chronic root pain but appears to be short-lived [9,10,11,12,13,14].

To date, there is no biochemical evidence of short or long-term side effects related to ozone administration and oxygen-ozone treatment has proven highly effective in relieving chronic low back pain and sciatica. Even more effective treatment is associated with the oral administration of ALA + PEA + myrrh.

Therefore, administration of the gas mixture associated with ALA + PEA + myrrh can be suggested as a first-choice treatment, to replace epidural steroid infiltration and avoid surgery.

Our findings confirm the results of Bonetti et al. [25] regarding the therapeutic efficacy and the duration of the effect of alpha lipoic acid + palmitoylethanolamide and myrrh associated with ozone therapy. Additionally, we found that the combined effect of this association is more effective than the sole treatment by steroid, allowing better pain control, especially in the later stages of the disease. This finding is in line with the literature suggesting oxygen-ozone is more effective than steroid infiltration [8], in the treatment of low back pain and provides evidence in favor of how APA + PEA and myrrh increase the efficacy of the sole oxygen-ozone therapy.

This study is not without limitations. The action of the biochemical factors at the basis of the outcome of these therapies may also be studied in more detail, in the future. This need is in line with what was discussed by De Sire et al. [23] in their recent review regarding oxygen-ozone therapy. We agree with the authors about the fact that it is necessary to create standardized procedures aimed at assessing the efficacy of treatments. Beyond research on oxygen-ozone alone, it will be useful to assess the efficacy of APA + PEA + myrrh + oxygen-ozone by creating research protocols that can be tested on wider samples.

It could be useful to repeat this research design by adding a comparison group without 800 mg/day of ALA + 600 mg/day of PEA + 200 mg/day of myrrh orally.

## Figures and Tables

**Figure 1 biomedicines-11-00073-f001:**
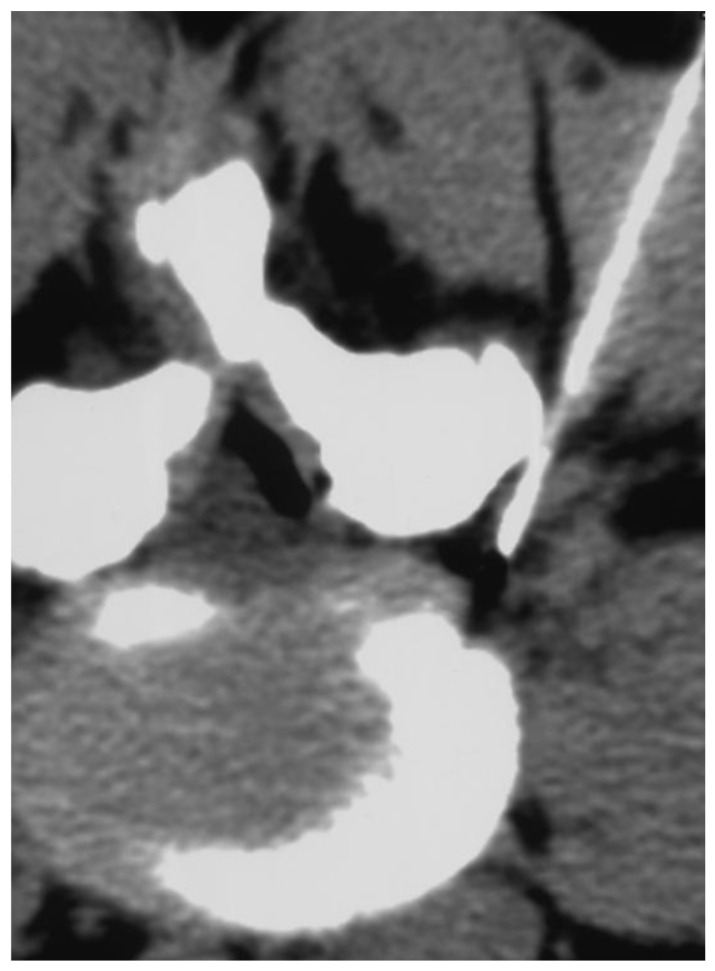
CT scan check of needle positioning before treatment by steroid infiltration.

**Figure 2 biomedicines-11-00073-f002:**
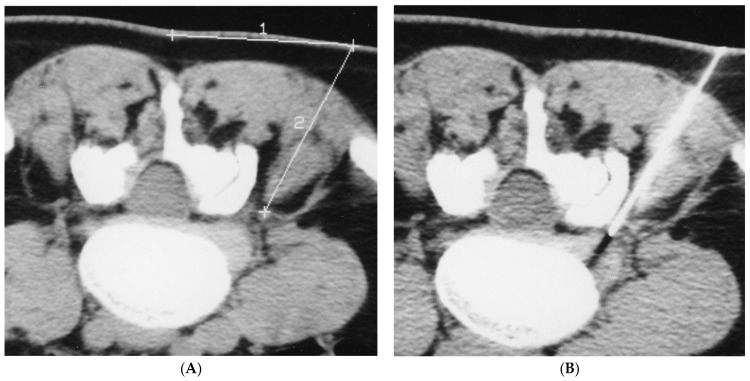
(**A**,**B**): The puncture site was identified by a CT scan and marked on the patient’s skin. The distance from this point to the foramen was subsequently measured. A 22 G Terumo needle (a 9 cm needle was typically used), was positioned 2–3 mm from the foraminal region, close to the ganglion of the affected nerve root. A CT scan was then repeated to check the correct needle placement.

**Figure 3 biomedicines-11-00073-f003:**
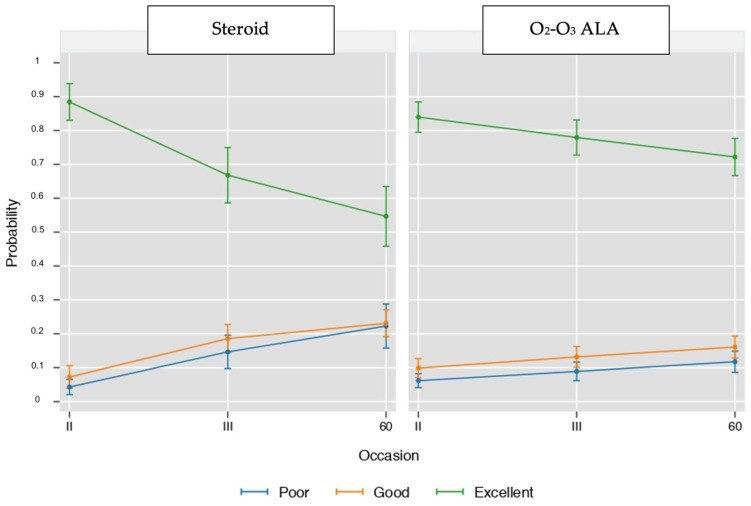
Time-related changes in the proportion of patients with poor, good, and excellent outcomes in the -ALA and steroid groups. Values are proportions and 95% cluster confidence intervals from ordinal logistic regression (see statistical analysis for details).

**Figure 4 biomedicines-11-00073-f004:**
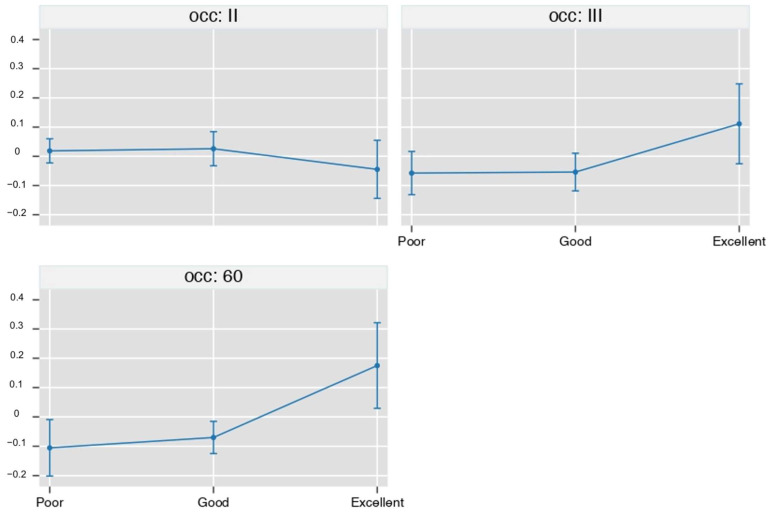
legend Between-group (O_2_-O_3_-ALA minus steroid) difference in the proportion of patients with a poor, good and excellent outcome during the study. Values are proportions and 95% cluster confidence intervals from ordinal logistic regression with Bonferroni correction for 9 contrasts. Points whose 95%CI do not cross the 0 line are significant at a *p*-level < 0.05 (see statistical analysis for details).

**Figure 5 biomedicines-11-00073-f005:**
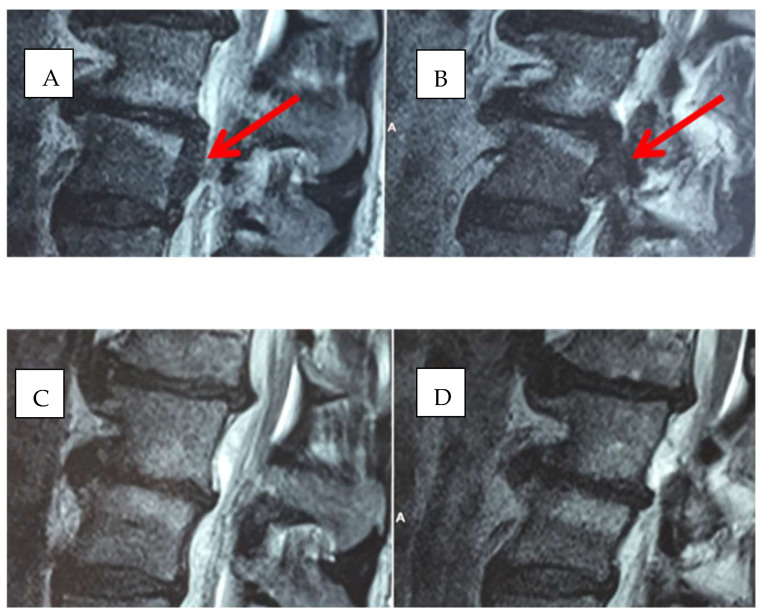
(**A**–**D**): Herniated disc in L4-L5 before (arrows) and after treatment with oxygen-ozone + ALA + PEA + Myrrh.

**Figure 6 biomedicines-11-00073-f006:**
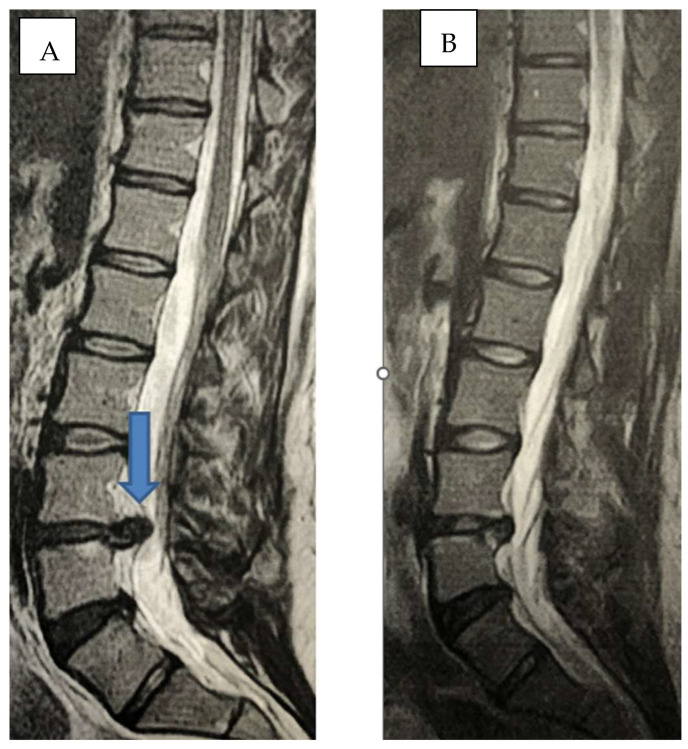
(**A**,**B**): Sagittal MR scan before (arrow) and after oxygen-ozone + ALA + PEA + Myrrh, herniated disc in L4-L5 (arrow), complete dehydration after treatment.

## Data Availability

Not applicable.

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
