# Peer review of "New Approach to Chronic Back Pain Treatment: A Case Control Study"

_biomedicines, 2022, doi:10.3390/biomedicines11010073_

Round 1

Reviewer 1 Report

Review of Biomedicines 2060012

New approach to chronic back pain treatment: a case control study.

Comments to the authors.

Abstract line 38    24 (18.8) reported   18.8%

Low back pain, with or without the involvement of the sciatic nerve, affects about 80% of the population at least once in their life and is the main cause of lost work days, with a strong impact on national health expenditure (1, 2).  This comment needs expansion since it is of central importance to this manuscript.

References 1 and 2 cited don't seem the most appropriate citations here

1. Andreula C.F., Simonetti L., De Santis F., Agati R., Ricci R., Leonardi M. Minimally Invasive Oxygen-Ozone Therapy for Lumbar 299 Disk Herniation. AJNR. 2003;24:996–1000. 300

2. Bonetti M., Zambello A., Leonardi M., Princiotta C. Herniated disks unchanged over time: Size reduced after oxygen-ozone 301 theapy. Interv. Neuroradiol. 2016;22:466–472.

Any of the following references would provide more relevant support to this introductory statement on the significance of low back pain.

1          Chou, R. Low back pain (chronic). . BMJ clinical evidence, 2010, 1116. 2010, 1116 (2010).

2          Briggs, A., Buchbinder, R. Back pain: a National Health Priority Area in Australia? . Med J Aust 190, 499-502 (2009).

3          Chou, R. In the clinic. Low back pain. . Ann Intern Med 150, ITC6-1 (2014).

4          Vos, T., Flaxman, AD, Naghavi, M, et al. . Years lived with disability (YLDs) for 1160 sequelae of 289 diseases and injuries 1990-2010: a systematic analysis for the Global Burden of Disease Study 2010. . Lancet 380, 2163-2196 (2010).

5          Ehrlich, G. Low back pain. . Bulletin of the World Health Organization 81, 671-676 (2003).

6          Maniadakis, N., Gray, A. . The economic burden of back pain in the UK. . Pain 84, 95-103 (2000).

7          Walker, B., Muller, R, Grant ,WD. Low back pain in Australian adults: the economic burden. . Asia Pac J Public Health 15, 79-87 (2003).

8          Yelin, E., Weinstein, S, King, T. . The burden of musculoskeletal diseases in the United States. . Semin Arthritis Rheum 46, 259-260 (2016).

9          Ravindra, V., Senglaub, SS, Rattani, A, et al. . Degenerative Lumbar Spine Disease: Estimating Global Incidence and Worldwide Volume.   . Global Spine J 8, 784-794 (2018).

10.       Hoy, D., March, L, Brooks, P, Blyth, F, et al  The global burden of low back pain: estimates from the Global Burden of Disease 2010 study. . Ann Rheum Dis 73, 968-974 (2014).

The mechanism of action of oxygen-ozone therapy-should be discussed more fully.  Please explain the term “mummification” better.

The authors make the statement “The possibility to orally combine alpha-lipoic acid (ALA) + palmitoylethanolamide (PEA) and myrrh with guided oxygen-ozone CT infiltration has been documented to be effective in further improving the clinical outcome. In a recent study, this association has been assessed and the authors concluded that it can be considered an excellent therapeutic solution, improving the good final clinical result with better control of symptoms, particularly in the first phase of the disease (25). However, in that study the necessity of further investigating the therapeutic effects of alpha-lipoic acid (ALA) + palmitoylethanolamide (PEA) and myrrh with guided oxygen-ozone CT infiltration was underlined.  In light of these considerations, we therefore decided to further evaluate the therapeutic results in a group of 246 patients treated with CT guided infiltrations of oxygen ozone associated with alpha-lipoic acid (ALA) + palmitoylethanolamide (PEA) and myrrh “.

It is unclear from this statement what the therapeutic basis of these interventions are.  This manuscript is an excellent opportunity to clarify this issue and the authors should have done so rather than leave this uncertain interpretation.

Line 85 Unlike other anti-oxidants that have full functionality in aqueous or fatty tissues, ALA exerts its anti-oxidant function in both water and lipids.  This property gives thioctic acid a broad spectrum of anti-oxidant action (33-36).  Please re-word more precisely.

Line 89.  MyrLiq® myrrh is a dry extract of Commiphora myrrh gummy resins with a high content of bioactive furanodienes, obtained through a patented extraction process, which  allows to preserve all the properties of the original raw material (36 – 38)”.  Please re-word.

The active pharmacologic agents in myrrh should be discussed more fully and their mechanism of action clarified to demonstrate their therapeutic value.

Line 95 “The best therapeutic result of oxygen-ozone therapy compared to the use of the steroid is already widely reported in the literature (8), however, to our knowledge, no other study aimed to assess and compare treatment with alpha-lipoic acid (ALA) + pal mitoylethanolamide (PEA) and myrrh with guided oxygen-ozone CT infiltration and  treatment with periradicular infiltrations of steroid”.  Please re-word more clearly.

Line 145 “There was total non-toxicity, as the titanium, teflon, glass and silicon, which are inert to the ozone, are in contact”-this could be worded better.

Line 166 “Such model employed the outcome “  please re-word

Line 204 “At short-term follow-up, most of the patients in both group had a complete remission of pain (80,5%% and 83.7% respectively of the patients treated with Depomedrol® respectively of the patients treated with the O2-O3 mixture associated with ALA+PEA+ Myrrh). Thus, there was good outcome regardless of the treatment administered, with effects not reaching statistical significance.” please re-word

Line 238 correct spelling “cytochines”  cytokines   “prostanglandins”  prostaglandins

Line 275 “effectful” efficacious?

Line 278  “This study is not without limitations. The clinical outcome of treatments may also be studied in more detail, in the future, as discussed by De Sire et al. (23) in their recent

review. We agree with the authors regarding the need of further research on assessing the best approach to improve the treatment effectiveness.”  Please re-word more clearly.

Author Response

Ferrara 24/11/20222

Dear Reviewer,

Thank you for your useful comments and suggestions about our study.

With this letter, we would like to underline how we made use of your suggestions to improve our article.

Abstract line 38    24 (18.8) reported   18.8%

We corrected this line.

Low back pain, with or without the involvement of the sciatic nerve, affects about 80% of the population at least once in their life and is the main cause of lost work days, with a strong impact on national health expenditure (1, 2).  This comment needs expansion since it is of central importance to this manuscript.

We expanded this part, referencing further data from literature: “Low back pain, with or without the involvement of the sciatic nerve, affects about the majority of the population at least once in their life and is the main cause of lost work days, with a strong impact on national health expenditure.

In particular, over 70% of people in developed countries develop low back pain at a certain point of their life. This condition usually improves within 2 weeks, however about 10% remain off work and about 20% experience pain after one year.

Between 15% and 45% of adults experience low back pain each year, and 5% of people refer to hospital care lamenting a new acute episode (1, 2)”.

References 1 and 2 cited don't seem the most appropriate citations here 

1. Andreula C.F., Simonetti L., De Santis F., Agati R., Ricci R., Leonardi M. Minimally Invasive Oxygen-Ozone Therapy for Lumbar 299 Disk Herniation. AJNR. 2003;24:996–1000. 300 

2. Bonetti M., Zambello A., Leonardi M., Princiotta C. Herniated disks unchanged over time: Size reduced after oxygen-ozone 301 theapy. Interv. Neuroradiol. 2016;22:466–472. 

Any of the following references would provide more relevant support to this introductory statement on the significance of low back pain.

1          Chou, R. Low back pain (chronic). . BMJ clinical evidence, 2010, 1116. 2010, 1116 (2010).

2          Briggs, A., Buchbinder, R. Back pain: a National Health Priority Area in Australia? . Med J Aust 190, 499-502 (2009).

3          Chou, R. In the clinic. Low back pain. . Ann Intern Med 150, ITC6-1 (2014).

4          Vos, T., Flaxman, AD, Naghavi, M, et al. . Years lived with disability (YLDs) for 1160 sequelae of 289 diseases and injuries 1990-2010: a systematic analysis for the Global Burden of Disease Study 2010. . Lancet 380, 2163-2196 (2010).

5          Ehrlich, G. Low back pain. . Bulletin of the World Health Organization 81, 671-676 (2003).

6          Maniadakis, N., Gray, A. . The economic burden of back pain in the UK. . Pain84, 95-103 (2000).

7          Walker, B., Muller, R, Grant ,WD. Low back pain in Australian adults: the economic burden. . Asia Pac J Public Health 15, 79-87 (2003).

8          Yelin, E., Weinstein, S, King, T. . The burden of musculoskeletal diseases in the United States. . Semin Arthritis Rheum 46, 259-260 (2016).

9          Ravindra, V., Senglaub, SS, Rattani, A, et al. . Degenerative Lumbar Spine Disease: Estimating Global Incidence and Worldwide Volume.   . Global Spine J 8, 784-794 (2018).

10.       Hoy, D., March, L, Brooks, P, Blyth, F, et al  The global burden of low back pain: estimates from the Global Burden of Disease 2010 study. . Ann Rheum Dis 73, 968-974 (2014).

We corrected the citations, replacing them with the following:

  1. Chou, R. Low back pain (chronic). . BMJ clinical evidence, 2010, 1116. 2010, 1116 (2010).
  2. Ehrlich, G. Low back pain. . Bulletin of the World Health Organization 81, 671-676 (2003).

The mechanism of action of oxygen-ozone therapy-should be discussed more fully.  Please explain the term “mummification” better.

We explained more deeply the action of oxygen-ozone on the intervertebral disk:

“It has been demonstrated that the mixture of oxygen and ozone gas reacts with the amino acids and carbohydrates that make up the nucleus pulposus of the intervertebral disc, allowing the volume reduction of the disc itself.

In the case of compression of the nerve roots following a discopathy, the oxygen ozone mixture can determine resolution to the clinical situation.

In particular, oxygen-ozone therapy produces various beneficial effects: it reduces the inflammatory process and related symptoms such as pain, promotes the elimination of non-vital tissue in the intervertebral disc thus decreasing its volume and compression of the spinal nerve root and promotes the elimination of harmful substances and toxins thanks to the stimulating action on the microcirculation (21–23)”

The authors make the statement “The possibility to orally combine alpha-lipoic acid (ALA) + palmitoylethanolamide (PEA) and myrrh with guided oxygen-ozone CT infiltration has been documented to be effective in further improving the clinical outcome. In a recent study, this association has been assessed and the authors concluded that it can be considered an excellent therapeutic solution, improving the good final clinical result with better control of symptoms, particularly in the first phase of the disease (25). However, in that study the necessity of further investigating the therapeutic effects of alpha-lipoic acid (ALA) + palmitoylethanolamide (PEA) and myrrh with guided oxygen-ozone CT infiltration was underlined.  In light of these considerations, we therefore decided to further evaluate the therapeutic results in a group of 246 patients treated with CT guided infiltrations of oxygen ozone associated with alpha-lipoic acid (ALA) + palmitoylethanolamide (PEA) and myrrh “.

It is unclear from this statement what the therapeutic basis of these interventions are.  This manuscript is an excellent opportunity to clarify this issue and the authors should have done so rather than leave this uncertain interpretation.

We explained in detail the basis of the interventions:

“What led us to associate the therapies in this study was the need to investigate in more depth the enhancement that they bring to the analgesic and anti-inflammatory actions of oxygen-ozone therapy alone, together with the greater maintenance of its clinical results over time, as suggested by the literature (25, 26)”.

Line 85 Unlike other anti-oxidants that have full functionality in aqueous or fatty tissues, ALA exerts its anti-oxidant function in both water and lipids.  This property gives thioctic acid a broad spectrum of anti-oxidant action (33-36).  Please re-word more precisely.

We corrected this part:

“Antioxidants can have functionality in aqueous or fatty tissues. ALA is an antioxidant that exerts its antioxidant function in both of these tissues. This property gives thioctic acid a broad spectrum of antioxidant action (34–37).”

Line 89.  “MyrLiq® myrrh is a dry extract of Commiphora myrrh gummy resins with a high content of bioactive furanodienes, obtained through a patented extraction process, which  allows to preserve all the properties of the original raw material (36 – 38)”.  Please re-word. 

The active pharmacologic agents in myrrh should be discussed more fully and their mechanism of action clarified to demonstrate their therapeutic value.

We corrected this part:

“Myrrh extract blocks proteins involved in the inflammatory process such as Cox and inhibits the formation of NO, ROS, TNF-α, PGE2, NF-kB and MAPK. Clinical studies indicate that myrrh extract is also able to induce significant improvements in oste-oarthritis (37 – 39).

MyrLiq® myrrh is a dry extract of Commiphora myrrh gummy resins with a high content of bioactive furanodienes.

This extract – obtained through a licensed process –allows to preserve all the properties of the original raw material.”

Line 95 “The best therapeutic result of oxygen-ozone therapy compared to the use of the steroid is already widely reported in the literature (8), however, to our knowledge, no other study aimed to assess and compare treatment with alpha-lipoic acid (ALA) + pal mitoylethanolamide (PEA) and myrrh with guided oxygen-ozone CT infiltration and  treatment with periradicular infiltrations of steroid”.  Please re-word more clearly.

We corrected this part:

“It has been widely reported in literature that oxygen-ozone is more effective than therapy through sole steroid administration (8).

However, to our knowledge, no other study aimed to assess and compare treatment with alpha-lipoic acid (ALA) + pal-mitoylethanolamide (PEA) + myrrh + guided oxygen-ozone CT infiltration and treatment with periradicular infiltrations of steroid.”

Line 145 “There was total non-toxicity, as the titanium, teflon, glass and silicon, which are inert to the ozone, are in contact”-this could be worded better.

We corrected this part

“Total non-toxicity has been guaranteed, since the titanium, teflon, glass and silicon necessary for therapeutic practice, which are inert to ozone”.

Line 166 “Such model employed the outcome “  please re-word

We corrected this line:

“This model considered the outcome…”

Line 204 “At short-term follow-up, most of the patients in both group had a complete remission of pain (80,5%% and 83.7% respectively of the patients treated with Depomedrol® respectively of the patients treated with the O2-O3 mixture associated with ALA+PEA+ Myrrh). Thus, there was good outcome regardless of the treatment administered, with effects not reaching statistical significance.” please re-word

We corrected this part:

“At short-term follow-up, most of the patients in both groups experienced a complete remission of pain (80.5% of patients treated with Depomedrol® and 83.7% of patients treated with the O2-O3 mixture associated with ALA+PEA+ Myrrh). Thus, a good result was recorded regardless of the administered treatment, with effects that did not reach statistical significance (Figures 3 and 4).”

Line 238 correct spelling “cytochines”  cytokines   “prostanglandins”  prostaglandins

We corrected this errors.

Line 275 “effectful” efficacious?

We corrected this error.

Line 278  “This study is not without limitations. The clinical outcome of treatments may also be studied in more detail, in the future, as discussed by De Sire et al. (23) in their recent review. We agree with the authors regarding the need of further research on assessing the best approach to improve the treatment effectiveness.”  Please re-word more clearly.

We corrected this part:

“This study is not without limitations. The action of the biochemical factors at the basis of the outcome of these therapies may also be studied in more detail, in the future.

This need is in line with what was discussed by De Sire et al. (23) in their recent review regard-ing oxygen-ozone therapy. We agree with the authors about the fact that it is necessary to create standardized procedures aimed at assessing the efficacy of treatments.

Beyond research on oxygen-ozone alone, it will be useful to assess the efficacy of APA + PEA + myrrh + oxygen-ozone by creating research protocols that can be tested on wider samples.

It could be useful to repeat this research design by adding a comparison group without 800 mg / day of ALA + 600 mg /  day of PEA + 200 mg / day of myrrh orally.”

We will be happy to answer any future questions.

Thank you for your kind attention.

Best regards

Prof. Dorina Lauritano

Reviewer 2 Report

Thank you for permitting me to review this manuscript 

In this study the authors prospectively assesed the effect of periradic-  ular infiltrations of oxygen-ozone under CT guide combined with 800 mg / day of ALA + 600 mg /  day of PEA + 200 mg / day of myrrh orally compared to f 176 patients with low back pain 26 treated with periradicular infiltrations of steroid

Better results were obtained in group one 

my main critic is that there is no comparison to a group without 800 mg / day of ALA + 600 mg /  day of PEA + 200 mg / day of myrrh orally

this should be discussed at least by a litterature search 

Line 84 please provide reference  PPR 

Figures should appear in the result section 

The authors should declare that their IRB waived the approval since the design was retrospective it is upon the IRB to state that not the authors 

In addition patients should be informed that their data can be published in a scientific study 

by the way it is not very clear if this study is retrospective or prospective observational   example line 110 enrollement is in a prospective study not in a retrospective study 

Author Response

Ferrara  24/11/2022

Dear Reviewer,

Thank you for your useful comments and suggestions about our study.

With this letter, we would like to underline how we made use of your suggestions to improve our article.

my main critic is that there is no comparison to a group without 800 mg / day of ALA + 600 mg /  day of PEA + 200 mg / day of myrrh orally

this should be discussed at least by a litterature search

We took into account this criticism, by adding it to the limitations of the article:

“This study is not without limitations. The action of the biochemical factors at the basis of the outcome of these therapies may also be studied in more detail, in the future.

This need is in line with what was discussed by De Sire et al. (23) in their recent review regard-ing oxygen-ozone therapy. We agree with the authors about the fact that it is necessary to create standardized procedures aimed at assessing the efficacy of treatments.

Beyond research on oxygen-ozone alone, it will be useful to assess the efficacy of APA + PEA + myrrh + oxygen-ozone by creating research protocols that can be tested on wider samples.

It could be useful to repeat this research design by adding a comparison group without 800 mg / day of ALA + 600 mg /  day of PEA + 200 mg / day of myrrh orally.”

Line 84 please provide reference  PPR

We corrected this part, referencing the corresponding papers.

Figures should appear in the result section

We added figures references in the result section, as suggested.

The authors should declare that their IRB waived the approval since the design was retrospective it is upon the IRB to state that not the authors

In addition patients should be informed that their data can be published in a scientific study

by the way it is not very clear if this study is retrospective or prospective observational   example line 110 enrollement is in a prospective study not in a retrospective study

We took into account this criticism, specifying our reasons and our research design in the following part:

“Institutional Review Board Statement: Ethics Committee or Institutional Review Board approval are not required for this manuscript, because treatments used (therapy with oxygen-ozone, alpha lipoic acid, palmitoylethanolamide, myrrh, steroid) cannot be classified as out of usual clinical limits of medical practice.

Furthermore, this study is comparative and retrospective, as clinical data used were performed before research analysis.”

In order to further specify our research design, we corrected line 110:

“Before enrolment, all patients had undergone a computed tomography (CT) or magnetic resonance (MR) scan.”

We will be happy to answer any future questions.

Thank you for your kind attention.

Best regards

Prof. Dorina Lauritano

Round 2

Reviewer 1 Report

The manuscript is now acceptable for publication

Author Response

Ferrara 01/12/2022

Dear Reviewer,

I thank you for accepting our manuscript.

Kind regards.

Prof. Dorina Lauritano

Reviewer 2 Report

the authors have adequately responded to most of my queries except for the etical concern

Author Response

Ferrara 01/12/2022

Dear Reviewer,

I thank you for accepting our revisions.

Kind regards.

Prof. Dorina Lauritano
